# The climate impact and land use of cultivated meat: Evaluating agricultural feedstock production

**Hanno Kossmann**[ID]**\*, Thorsten Moess, Peter Breunig**

Faculty of Agriculture, Food and Nutrition, Weihenstephan-Triesdorf University of Applied Sciences, Weidenbach, Germany

\* hanno.kossmann@hswt.de

**Data Availability Statement:** All relevant data are within the manuscript.

## Abstract

As global demand for meat continues to rise, alternative and sustainable methods of production are being explored. Cultivated meat (CM) is one such alternative that holds potential for sustainable production with less environmental impact. This study develops an approach to evaluate CM production based on agricultural feedstock. The specific objectives are to determine the minimum land area required to produce a certain amount of cell medium—feedstock for CM production—on agricultural land and to identify potential future land use scenarios assuming that the macro components of the cell medium are solely produced from common agricultural crops in southern Germany. A linear programming model was developed to analyze four different scenarios of CM production, considering factors such as crop rotation, nutrient sourcing, and solar energy use. The results indicate that CM production using plants as raw material for the cell medium cannot improve land use efficiency substantially compared to conventional pork production. Extraction methods, crop choice, and energy sources will strongly influence future pathways for CM. We also find that there is no substantial benefit from CM in terms of climate change mitigation when feedstock is sourced solely from plants. This study provides valuable insights into the limitations of using agricultural feedstock for sustainable CM production. The findings suggest that future research should focus on optimizing the land use efficiency of CM. This includes exploring alternatives such as sourcing cell media from precision fermentation instead of relying solely on crops, and utilizing upcycling possibilities.

## 1 Introduction

Human diets have prominently incorporated meat for millennia, serving as a cornerstone for nutritional needs and livelihoods throughout the world [1–3]. Despite declining meat consumption in high-income countries, forecasts indicate a ballooning demand for meat in forthcoming decades due to global population surges and increasing incomes [4, 5]. Plant-based foods are not always available or affordable and livestock is an important source of income to poor households [6]. Globally meat is instrumental for food security and livelihoods [7, 8].

**Funding:** Funding for the authors' research position, which supported the completion of this study, was provided by the Bavarian State Ministry of Science and the Arts as part of the High-Tech Agenda Bavaria. This article was additionally funded by the Open Access Publication Fund of Weihenstephan-Triesdorf University of Applied Sciences. The funders had no role in the study design, data collection and analysis, decision to publish, or preparation of the manuscript.

**Competing interests:** The authors have declared that no competing interests exist.

But the consumption of protein-rich foods is marked by their substantial land and resource demands [9–12]. In addition to complex concerns surrounding animal welfare and ethical considerations [13–15], the resource utilization of animal-sourced foods has become a focal point in both everyday discourse and academic research. Increasing consumption rates in lower-income countries and the Global South also lead to dependencies and geographically detached environmental influences due to imported animal feed [16, 17].

Pork is the most consumed meat globally, and its production alone covers more than 5% of all croplands worldwide. The feed conversion ratio and environmental impact of pig farming are highly dependent on effective management practices and the quality of feed used. Proficient management can significantly enhance the sustainability of pig farming operations [18]. Based on feed conversion ratios, pork production efficiency lies between that of chicken and beef [19, 20]. Life cycle assessments (LCAs) accentuate the extensive environmental footprint of pork production, encompassing factors like greenhouse gas emissions, land and energy use [21].

Feed production and mixture, which influence land use, as well as waste management, which affect methane emissions and nitrate pollution, are the most critical factors impacting the environmental footprint of pork production [22]. These environmental concerns, also linked to meat production in general, have amplified the quest for sustainable alternatives, especially those with a rich protein content [23–25].

A subset of these innovations are animal proteins made from cell cultivation, which directly mimic traditional animal products in flavor, appearance, and composition. Cultivated meat (CM) represents one of those technologies [26–28]. Using in-vitro techniques, animal cells are cultivated in a controlled setting, obviating animal rearing, slaughter, and thereby animal welfare issues [29–31]. Notably, the CM domain witnessed significant strides with the unveiling of the first CM burger in 2013 [32].

Consumers are generally open to eating CM [33, 34]. Especially ethical advantages in terms of environmental sustainability and animal welfare are influencing consumers' willingness to try and eat CM [35]. On the other hand, perceived risks and unnaturalness make consumers critical [36].

Existing LCAs on CM present a spectrum of conclusions, from overtly skeptical [37–39] to decidedly optimistic [40–42]. While CM emanates promise as a greener substitute for traditional meat, the sector remains nascent with pronounced information voids [23, 30, 43]. Particularly, there is a dearth of clarity on how emerging countries, where meat demand is rising, can develop the necessary capabilities to integrate CM technology, as most startups are based in developed countries [44].

Given the elevated demand for meat–and possibly CM–substantial input commodities will be required for the nourishment of CM cells. Consequently, agricultural production could play a pivotal role in this supply chain [45]. Central to CM's efficacy is cell media, which is a composite vital for in-vitro cellular growth [41, 46]. The cost-effective synthesis of cell media is pivotal for ensuring efficiency in resource and monetary terms. The cost of cell media which is 55%–95% of CM's total cost, is mainly caused by cytokines and transferrine production [47, 48]. The cause of these high costs is mainly the lack of industry scaling [49]. Current research posits agricultural crops, predominantly maize and soy, as feasible crop-based nutrient sources for CM's media [40, 41, 50]. But some parts of the feedstock are also added by precision fermentation microbial or chemical production in the case of Sinke et al., 2023 [41] for instance 25%, while 75% from soy hydrolysate. The production of cell culture medium ingredients has high influence on the environmental impact of CM [42]. Contrary to Sinke et al. (2023) [41], Tuomisto et al. (2022) [42] further recommend exploring amino acid production using hydrolysates from legumes among other things.

Besides glucose and amino acids as main components of the cell media, electrical energy is another major input in CM production, since there is no complete organism regulating cell temperature like in animals [41].

So far, there is no research that investigates optimization opportunities in land use for producing the macronutrients (glucose and amino acids) and renewable electricity required to enable CM production.

This study evaluates the use of agricultural feedstock and land use for cultivated pork production in terms of macro nutrients and energy demand. We use linear programming to optimize CM output per land area based on the cell nutrient and energy demand as well as crop yield and crop rotation restrictions. The optimized land use demand for CM is compared to land use requirements of pork production. In addition, we quantify the climate impact of land use through carbon opportunity cost.

To apply this approach to a specific region we use yields and crop rotation restrictions of southern Germany. This region has the environmental conditions to grow a large variety of crops with relatively high yields and robust data.

This paper unfolds as follows: Section 2 delves into the background of CM, charting its technological background, environmental implications, land use, economic viability, and impact on agriculture. Section 3 elucidates our methodology—encompassing data acquisition, analytical techniques, and the structure of linear optimization. Section 4 showcases our results, which we subsequently discuss. The paper culminates with reflections on our findings and an outline for future research trajectories.

## 2. Background of cultivated meat

Technical and environmental aspects of CM production, innovative approaches, regulatory challenges, and the importance of transparent and accurate information for consumer acceptance will be described in this section. CM research is in its early stages, focusing on promising narratives, understanding consumer behavior, and exploring technical and regulatory feasibility [23, 32, 36, 47]. CM provides an alternative to conventional animal farming, potentially leading to a more sustainable and ethical meat production system by eliminating the need for animal slaughter and rearing. CM's market introduction could significantly impact monogastric animal production by producing fewer by-products and not using land unsuitable for human food production [29].

Around the world, many startups have ventured into the CM industry, securing significant funding [51]. However, for successful market introduction, it is crucial to address production techniques, scaling challenges, environmental concerns, regulatory hurdles, and consumer acceptance [23, 25, 41, 52, 53]. Several technical challenges hinder CM from becoming a viable alternative to conventional meat. A primary concern is the industry's reliance on fetal bovine serum for cell culture, which is ethically contentious and not scalable. Efforts are underway to develop serum-free alternatives [32, 54]. Criticism states that Moore's law, speaking of an almost linear cost to doubling output relation, is not applicable to biological systems [12]. Another challenge is replicating the texture and flavor of traditional meat, as current CM production results in disorganized muscle fibers that lack the complex structure of conventional meat cuts [55]. Additionally, assessing the nutritional quality of CM, particularly its micronutrient composition and overall health implications, requires further research and technological advancement [54, 55].

From a regulatory standpoint, CM is being managed differently across the globe. Singapore leads with early approvals and frequent updates. China and India are in the process of developing their frameworks. Japan, South Korea, Australia-New Zealand, and the EU regulate under

existing novel food guidelines. Post-Brexit, the UK is creating specific regulations, and the US is coordinating between the FDA and USDA. Canada and Brazil are also progressing in their regulatory approaches [56].

## 2.1 Technical background of CM production

CM production encompasses various stages: cell isolation, cell expansion, differentiation, bioreactor utilization, and harvesting. Initiated by extracting muscle or fat cells from animal tissues, these cells are nurtured in a nutrient-dense culture medium. Following sufficient proliferation, these cells undergo differentiation into muscle cells. Subsequently, this tissue is matured in a bioreactor until it reaches an optimal state for harvesting and processing into meat products. Preliminary prototypes spanning beef, pork, and poultry highlight the potential of this approach [37, 50, 56].

Culture media, instrumental in in vitro meat production, present economic and technical challenges. Current formulations, although general, necessitate tailoring for specific cellular metabolic states. Although fetal bovine serum, a meat industry byproduct, is commonly used in cell culture media, serum-free alternatives are under investigation to alleviate associated ethical concerns [47, 57]. Moreover, innovations are imperative in cellular modulation. These could be reached by physical and chemical conditioning, optimized substrate elasticity, and nutrient diffusion in expansive tissues [29, 32, 58].

Formulating cell media entails sourcing from diverse origins and processes. Core components include glucose, amino acids, vitamins, and trace elements crucial for optimized cell growth and yields. O'Neill (2021) [46] suggests incorporating renewable feedstocks like lignocellulosic hydrolysates into media formulations for environmental sustainability. Vergeer's & Risner's LCAs emphasizes the role of eco-friendly practices in media formulation, touching on issues such as the environmental ramifications of antibiotics and the strain from glucose production [58, 59]. Research supports cell media derivation from plants like soy and maize for sugars and amino acid extraction, alongside utilization of precision fermentation or biotechnological processes [40, 41, 50, 60–62]. This study accentuates plant-based media composition, considering its implications on agriculture and land use.

Cell metabolism encapsulates the myriad chemical reactions essential for cellular sustenance and growth, including nutrient breakdown, molecular synthesis, and waste elimination. The vital components for these reactions range from glucose, amino acids, and vitamins to oxygen and growth factors like hormones. To evaluate the demand of cells for efficient growth of CM, we assumed the following metabolism reaction taken from Humbird (2021) [37]:

$$
\begin{aligned}
0.147\ Glc &+ 0.378\ O2 + 0.007\ Arg + 0.004\ Cys + 0.022\ Gln + 0.003\ His + 0.007\ Ile \\
&+ 0.010\ Lys + 0.002\ Met + 0.005\ Phe + 0.009\ Thr + 0.002\ Trp + 0.005\ Tyr + 0.010\ Val \\
&+ 0.013\ Ala + 0.006\ Asn + 0.008\ Asp + 0.011\ Gly + 0.011\ Leu + 0.007\ Pro + 0.010\ Ser \\
&\rightarrow
\end{aligned}
\tag{1}
$$

$$
DCMa + 0.005\ Glu + 0.004\ NH3 + 0.041\ Lac + 0.455\ CO2 + 0.613\ H2O
$$

In the fed-batch process as assumed in Humbird (2021) [37], cells are cultivated in a nutrient-rich medium, replenished periodically. Contrary to perfusion cultures, where nutrients are provided and waste is removed continuously, with the fed-batch process the medium is depleted as cells reach certain densities. The fed-batch technique is cheaper, but cell growth may be limited to a certain extent [37, 43].

For CM to be widely accessible, scalable production is vital. Humbird (2021) [37] expounds on the challenges, emphasizing bioreactor issues during fed-batch operations. Compared to

microorganisms, scaling is complex due to slow animal cell growth rates combined with shear force-induced cell damages, metabolic inhibition, and aseptic maintenance challenges. Perfusion technology offers a potential solution, although it limits bioreactor sizes. Elevated hygiene demands, combined with the need for richer nutrient mediums for animal cells, especially stem cells, further amplify the challenges [37].

## 2.2 Environmental implications and challenges of CM LCAs

Finally, the environmental impact of CM will be crucial to its success [31]. Evaluating the environmental impact of food products requires examining land use efficiency [63] and understanding the carbon opportunity costs associated with land use [64].

In this study, we employ land use as a metric to explore the potential environmental advantages of CM. While a singular focus on land use does not fully encompass all environmental ramifications of meat production, it remains an informative, albeit partial, metric for this analysis due to two main reasons. First, land use encompasses a significant majority of the environmental costs associated with pork production, especially soy cultivation for feed production [65–67]. Second, land use serves as a potent indicator of the biodiversity impacts tied to various agricultural products [68].

LCAs for CM have been conducted to evaluate the environmental impacts of this technology [38, 41, 42, 43, 50, 59, 62, 69]. These assessments typically cover the entire product lifecycle, from raw material production and energy use to cell cultivation, processing, and packaging. LCAs of CM evaluate various environmental impacts, including emissions (e.g., CO2), acidification, eutrophication, and health metrics (carcinogens, ecotoxicity, toxicity). They also assess resource use, such as energy demand, fossil resource scarcity, and land use, providing a comprehensive view of CM's sustainability.

In summary, existing LCAs indicate that CM represents a technological shift in meat production, using more energy but requiring less and different feed than conventional meat. CM minimizes emissions and performs well on several environmental indicators, but its sustainability depends on renewable energy and efficient resource management. Despite some uncertainties, CM shows promise for a lower carbon footprint and reduced environmental impact, particularly as energy systems decarbonize. Studies highlight CM's lower biotic resource use and potential environmental benefits, although energy use remains a critical challenge, emphasizing the need for decarbonized energy sources and innovative solutions [42]. Sinke et al. (2023) [41] see energy and culture medium as the main two future challenges. Energy demand is high due to bioreactor systems and scales, calculation methods that are conservative, heat dynamics at larger scales, and variability in heating/cooling efficiency; however, there is a lack of large-scale empirical data on the energy demand of producing CM at large scales [42].

Several LCAs examine different methodologies for producing cell media for CM, emphasizing various biological pathways. Some focus on using crops as the base material for cell media [41, 42, 50], while others explore using bacteria or alternative microorganisms [61, 70]. Consequently, the specific composition and production processes of cell media significantly affect resource utilization in CM production.

Recent LCAs report that the land use of CM is consistently lower than that of conventional meat, with values ranging from 0.4 to 6.9 $m^2$/kg meat [41, 42, 43, 50, 61]. This variation is due to differences in raw material selection, databases, and case study locations. This study uses a specific example that accounts for crop rotation restrictions and local yields. Energy use and culture medium directly impact land use, considering feed and sustainable energy production. Most LCAs use agricultural modeling databases to calculate land use for CM. However, incorporating agricultural realities and modeling crop rotation effects pose challenges [42, 43]. We

employed a case study approach to compute land use and carbon opportunity costs for CM production in southern Germany, using on-site generated feed, local yields and nutrient contents.

## 2.3. Economic viability and impacts on the meat industry and farming communities

The economic viability of cultured meat (CM) is critical for its scalability and market adoption. Advances in large-scale CM production are closely tied to developments in the biomedical field, particularly in tissue engineering. Key innovations include the development of specialized cell lines, plant-based cell media, advanced scaffolding materials, and efficient bioreactors. These innovations are foundational to the growth and competitiveness of the CM sector, and their ongoing development is essential for reducing production costs and improving product quality [71, 72]. Projected production costs vary significantly depending on the process, ranging from estimates of $5.50/kg [40] to $25/kg [37]. However, numerous assumptions make these values approximate and subject to considerable uncertainty. These scenarios require much of the necessary infrastructure that does not yet exist. Furthermore, the successful integration of CM into the global value chain requires not only technological advancements but also strong business structuring, market positioning capabilities, and robust relationships with stakeholders. Addressing these multifaceted challenges and opportunities is crucial for the widespread adoption of CM, particularly in emerging countries where supportive policies and infrastructure can play a pivotal role in fostering industry growth [73].

Historically, the meat industry and farming communities have been central to the global food system, impacting cultural, economic, and societal frameworks. Their contribution transcends direct agricultural activities, providing economic stability to many communities [19]. The emergence of CM could redefine these traditional roles, potentially diminishing the centrality of conventional livestock farming as technological advancements in CM become more pronounced [32, 71].

However, the integration of CM into existing agricultural frameworks might offer new opportunities. For instance, CM production could become decentralized, with potential for implementation at the farm level, thereby including traditional farmers in the emerging CM industry [73]. Also new and higher-skilled jobs are expected to be created [74].

The economic landscape may shift as CM becomes more established. Traditional meat producers may face challenges in maintaining revenue [74], whereas CM offers new economic avenues, particularly for those willing to diversify into this new area [75]. For example, farmers could utilize their expertise to contribute to the CM supply chain, such as in the production of cell media, demonstrating the adaptability and resilience of farming communities [45, 74].

Despite these opportunities, there are barriers to the widespread adoption of CM. Many farming communities, steeped in generational traditions, may view CM with skepticism or resistance, questioning its authenticity and the impact on traditional farming practices. These observations could also be made for veterinarians and animal scientists in Brazil [74]. Integration with CM could require significant investments in new equipment, knowledge, and infrastructure tailored to this innovative industry [30, 76]. While private funds have been used most recently, scientists and other interest groups are calling for additional public funding [49].

Morais-da-Silva et al. (2022) [74] in turn saw from interviews of alternative protein experts more positive reflections and opportunities outlined by the experts. The experts saw the potential to create new, higher-skilled jobs, particularly in technical and management fields. However, the transition may disproportionately affect animal farmers, especially in a rapid change

scenario, with varying levels of optimism across regions, most notably higher in Brazil than in Europe.

Furthermore, understanding the future role of agriculture in CM production necessitates a comprehensive analysis of resource utilization and the sustainability of CM compared to traditional meat. This includes evaluating the land use requirements of CM to ascertain its environmental benefits and value-added potential in the agricultural sector.

## 3. Methods and scenarios

### 3.1 Modeling structure

Our model uses linear programming to maximize CM production within a 100-hectare reference area, incorporating crop rotation and nutrient constraints specific to southern Germany. Linear programming is a mathematical strategy frequently used across various disciplines for optimization purposes [77]. In agriculture, it is often employed to optimize resource utilization, encompassing factors such as land and crop rotation optimization [78–81].

In the context of our research, we employed a linear programming model with several specific components in mind. Primarily, we defined an objective function that represents our optimization goal: maximizing the production of CM per 100 hectares of land. In the modeling structure, the 100 hectares serve as a land area reference scale for comparison. The results are expressed in kilograms of meat per square meter. Our model also incorporates constraints reflecting the crop restrictions specific to southern Germany. We focus on southern Germany since this region has environmental conditions that allow the production of a wide variety of crops with relatively high yields (Table 1). These constraints include crop rotation practices, solar energy use, and meeting CM nutrient requirements. Although future methods may produce some amino acids, we focused on a closed, plant-based system. By integrating these components, the linear programming model utilized in this study is designed to identify the optimal land use strategies. The ultimate objective is to maximize the production of CM within the confines of limited land availability. Hence, the model offers valuable insights into efficient land practices and optimal crop mixtures for achieving the highest possible output of CM.

**Table 1. Average crop yields in southern Germany, Germany, the world, and continents in 2021.**

| Crop | Southern Germany (Bavaria) (t/ha)[a] | Germany (t/ha)[b] | Africa (t/ha)[b] | Americas (t/ha)[b] | Asia (t/ha)[b] | Europe (t/ha)[b] | World (t/ha)[b] |
|---|---|---|---|---|---|---|---|
| Barley | 6.75 | 6.76 | 1.77 | 2.87 | 1.59 | 3.96 | 2.97 |
| Lupine | 2.88 | 1.84 | 0.70 | 1.61 | 2.25 | 1.58 | 1.40 |
| Maize | 10.52 | 10.36 | 2.27 | 7.80 | 5.58 | 7.20 | 5.87 |
| Oats | 4.45 | 4.32 | 1.30 | 2.38 | 2.03 | 2.52 | 2.36 |
| Peas (general) | 2.99 | 3.06 | 1.19 | 1.47 | 1.45 | 2.32 | 1.76 |
| Potatoes | 42.26 | 43.79 | 15.02 | 29.44 | 19.15 | 23.62 | 20.74 |
| Rapeseed | 3.57 | 3.50 | 1.89 | 1.53 | 1.72 | 2.81 | 1.93 |
| Rye | 5.52 | 5.27 | 1.87 | 2.43 | 2.29 | 3.23 | 3.05 |
| Soybean | 3.17 | 3.11 | 1.44 | 3.27 | 1.43 | 2.09 | 2.86 |
| Sunflower | 2.70 | 2.60 | 1.05 | 1.92 | 2.02 | 2.06 | 1.97 |
| Triticale | 4.88 | 5.81 | 2.30 | 2.80 | 2.06 | 4.12 | 3.87 |
| Wheat | 7.54 | 7.30 | 3.03 | 2.83 | 3.39 | 4.28 | 3.49 |

[a]LfL, 2023 [82]

[b]FAOSTAT, 2023 [83]

Note: t/ha = tons per hectare.

Crop rotation restrictions are detailed in Table 2. For grain legumes, total acreage cannot exceed 20% of the farm area, with specific caps for individual crops: lupine, field bean, and pea are each limited to 15% [84].

The objective to be maximized is the expansion of the activity fresh mass production (referred to as the target cell). To achieve this goal, a solver in the model is tasked with adjusting the expansion of all activities, including that of fresh mass production. This adjustment must be carried out while adhering to certain constraints. These constraints include the full utilization of land area capacity, crop rotation restrictions (mentioned below), and the required nutrient supply for agricultural crops together with the corresponding expansion of production procedures.

## 3.2 Model specification

This section specifies the objective and constraints of the linear programming model to optimize nutrient and land use for CM production. We developed a static linear programming model to evaluate the agricultural feedstock production for CM in southern Germany, applying it to a 100-hectare land area reference. The model aims to maximize CM output while adhering to agricultural constraints like yields and crop rotation restrictions. The focus is on using 100 hectares to produce the macro nutrients (glucose and amino acids) and energy required for CM production.

The model's objective function and constraints are formulated as detailed below:

Objective Function:

$$\text{Maximize } M = p^T x \tag{2}$$

Where:

- $M$ is the total production of CM medium (targeting maximization).

- $x$ is the vector of decision variables, representing the activities including different crops and crop rotation, as well as the potential use of solar panels for energy production.

- $p$ is the vector of production quantities per unit of activity, namely the quantities of glucose and single amino acids produced per unit of crop or solar energy generation activity.

- $T$ denotes the transpose of the vector $p$.

Constraints:

Nutrient Recipe Constraints:

These constraints ensure that the mix of inputs used matches the specific nutritional requirements for CM production:

$$\sum_j a_{ij} \cdot x_j \cdot y_j \geq b_j \cdot M \tag{3}$$

where:

- $a_{ij}$ represents the amount of amino acids and glucose content $i$ provided by crop $j$.

- $x_j$ is the amount land use for a specific crop or for energy production $j$.

- $y_j$ is the yield per ha of crop or energy production

- $b_i$ is the required amount of input $i$ per unit of CM medium, defining the feedstock recipe for CM.

The goal of the model is to find the optimal combination of activities $x$ that maximizes the total production of CM medium $M$, subject to the constraints of available resources, which is land and crop rotation restrictions.

Crop Rotation and Land Use Constraints:

Account for crop rotation and land use practices:

$$x_j \leq L \cdot CR_j \tag{4}$$

where:

- $x_j$ is the land used for crop $j$.

- $L$ is the total available land.

- $CR_j$ is the crop rotation factor for crop $j$, representing the maximum proportion of land that can be used for this crop to ensure sustainable rotation.

Account for total land use constraint:

$$\sum_{j \in Crops} x_j \leq L \tag{5}$$

The model aims to identify the optimal combination of activities $x$ that maximizes the total production of CM medium $M$, while complying with the available resources and crop rotation restrictions. The model's outcome serves to calculate and compare the area of land (in square meters) required per kilogram of produced CM compared to pork meat, or conversely, the yield of CM compared to pork meat (in kilograms) per square meter of land. This provides a comparable metric for assessing the efficiency and sustainability of the CM production process in southern Germany as the case study region.

We used Microsoft Excel's Solver Add-In for linear programming. Solver's optimization function balanced CM production nutrient requirements with crop area allocation, considering crop rotation constraints.

This linear programming approach provided insights into which crops were most limiting and which nutrients were in surplus or deficit.

## 3.3 Technical inventory and background: Linear programming for CM

Our study applies strict crop rotation restrictions to mirror realistic farming practices and ensure sustainable soil management in southern Germany. Regardless of farm structure or land ownership, these restrictions limit the share of each crop to maintain long-term yield levels. We examined protein plants (soybean, lupine, field bean, pea, winter rapeseed, and sunflower) and starch plants (grain maize, winter wheat, rye, triticale, winter barley, oat, and potato). Table 1 provides crop yield statistics, showing that southern Germany's yields are above the world average [83].

Soybeans are traditionally grown in warm regions, but advances in plant breeding and agricultural technology have made it possible to grow adapted varieties in more northern regions. In southern Germany, soybean farming increases crop diversity and reduces reliance on imported soybeans for livestock feed [85].

Challenges such as pest and disease risks and limited domestic market demand must be considered. We applied a soybean restriction to a maximum of 50% of the land area. A two-year maize-soy rotation system, proven feasible in Canada and the US Midwest [86–88], is assumed for southern Germany.

Summer crops are limited to no more than two-thirds of the land area. Grain maize is capped at 50%, while sunflower and potato each have a 20% cap. Winter crops also follow the two-thirds limit for the total farm area, with specific restrictions: wheat at 33.33%, rye at 50%, triticale at 33.33%, winter barley at 33.33%, oats at 25%, and rapeseed at 20%. These rotation limits are based on data from Diercks & Heitefuss (1990) [89], Kolbe (2021) [84], and Reckling et al. (2016) [90] (Table 2).

**Table 2. Crop rotation restrictions assumed in the linear programming model.**

| Crop category | Category limit (%) | Individual crop restrictions (%) |
|---|---|---|
| Grain legumes | ≤ 20% (without soy) | Soybean ≤ 50%, lupine ≤ 15%, field bean ≤ 15%, pea ≤ 15% |
| | ≤ 50% (with soy) | |
| Summer crops | ≤ 67.77% | Grain maize ≤ 50%, sunflower ≤ 20%, potato ≤ 20% |
| Winter crops | ≤ 67.77% | Wheat ≤ 33.33%, rye ≤ 50%, triticale ≤ 33.33%, winter barley ≤ 33.33%, oats ≤ 25%, rapeseed ≤ 20% |
| Foliage crops | ≥ 25% | Soybean, lupine, field bean, forage pea, rapeseed, sunflower |
| Rapeseed + Sunflower | ≤ 33.33% | Rapeseed, sunflower |
| Cereals | ≤ 67.77% | Wheat, rye, triticale, barley, oats |

Our study uses data on agricultural crops in Germany and feedstuff analyses to determine amino acid patterns and glucose contents [91, 92]. The INRAE-CIRAD-AFZ Feed Tables—a comprehensive reference for feed composition and nutritional values developed by French research institutions INRAE, CIRAD, and AFZ—are used for glucose quantity specifications [93]. Glucose content is derived from starch and sugar, however, the database does not provide detailed information on the monosaccharide composition of these fractions. For energy and oil crops, starch was considered primarily as D-glucose polymers, allowing near-complete conversion to glucose during hydrolysis. For grain legumes, only half of the starch and sugar fraction was considered convertible to glucose due to non-starch polysaccharides and oligosaccharides. Based on Humbird (2021) [37], an overall amino acid yield of 70% after extraction from crops was assumed. Due to feed analyses capturing glutamine and asparagine with glutamic acid and aspartic acid, these amino acids were considered collectively (Table 3). To quantify nutrient delivery, we used average yields from 2017–2021 calculated by the Bavarian State Research Center for Agriculture [82]. Nutrient yield per hectare was calculated by multiplying crop nutrient contents (%) with per-hectare yields (t FM/ha), considering dry matter contents of the feedstuffs [83, 94].

The energy demand for CM production was sourced from Vergeer et al. (2021) [40], Mattick et al. (2015) [50], and Tuomisto et al. (2014) [95]. Information on solar panel efficiency came from Böhm et al. (2022) [96]. Crop productivity data was obtained from the Bavarian State Research Center for Agriculture (LfL, 2023) [82].

Finally, we assume that all raw materials, including soy, are produced on-site by the farm, which is set at 100 hectares. The linear programming model then balances the nutrient requirement (kg/kg CM) with the nutrient delivery (kg/ha) for CM production.

A specific correction is applied that takes into account that the LfL data relates to 14% or 9% moisture (equivalent to 91% or 86% dry matter). This results in an adjustment of hectare yields based on the dry matter content of each analyzed feed material. Here is an example, in the case of soybeans.

Dry matter content per analysis:

Yield of 3.17 t/ha at 14% and dry matter following feed analysis of 89.5% results in a calculation of 3.17 t/ha x 0.86/0.895, or approximately 3.05 t/ha.

This corrected yield per hectare is then multiplied by nutrient content percentages, the nutrient extraction factor, and a conversion factor of 1,000 kg/t. This calculation gives the nutrient delivery in pure nutrient kilograms per hectare for each crop type. Table 3 presents the nutrient requirements for CM. The nutrient delivery is directly correlated with the nutrient requirement, expressed in kg/kg of CM, ensuring a consistent comparison of values in the analysis.

**Table 3. Relative nutrient requirements for cell mass production.**

|  | g/g CM FM stoichiometric demand[a] |
|---|---|
| Glucose | 0.362 |
| Arginine | 0.016 |
| Cysteine | 0.005 |
| Glutamine | 0.044 |
| Histidine | 0.006 |
| Isoleucine | 0.012 |
| Lysine | 0.020 |
| Mehionin | 0.004 |
| Phenylalanin | 0.011 |
| Threonine | 0.014 |
| Tryptophan | 0.004 |
| Tyrosine | 0.012 |
| Valine | 0.016 |
| Alanine | 0.016 |
| Asparagine | 0.011 |
| Aspartic acid | 0.014 |
| Glycine | 0.011 |
| Leucine | 0.019 |
| Proline | 0.010 |
| Serine | 0.015 |
| Glutamic acid + Glutamine | 0.044 |
| Asparagine + Aspartic acid | 0.025 |

[a]Humbird, (2021) [37].

Note: g/g CM = gram per gram cultivated meat; FM = fresh mass.

For energy and oil plants, it is presumed that starch is present as amylopectin (approximately 70–80%) and amylose (20–30%), both of which are polymers of the monosaccharide D-Glucose (primarily α-1,4-glycosidic bonds). It is approximated that starch or disaccharides can be broken down completely and utilized as glucose. This assumption is justified by the compensatory effect between any decomposition losses and the mass increase of the glucose fraction caused by the addition of $H_2O$ (H+ and OH- adaption) during hydrolysis [37].

With grain legumes, the case arises where some non-starch polysaccharides or oligosaccharides are analytically captured in the starch fraction, which, besides glucose, have up to 50% other monosaccharides (Mannose, galactose, and fructose) as basic building blocks. There are also considerable varietal differences [91, 92]. Therefore, it's roughly assumed that for these plants, 50% of the analytical sugar and starch fraction can be present as, or can be converted to glucose. Humbird (2021) [37] presumes a protein solubility of 88% and a hydrolysis conversion of 80% for hydrolysate from soybean meal, which corresponds to an overall amino acid yield of 70.4%. As stated before, an amino acid yield of 70% is assumed for all considered agricultural crops in this study.

The data on each crop's relative nutrient supply, as detailed in Table 4, serves as a foundational parameter for our calculations and is thus essential for determining the nutrient yield per crop. Our analysis distinctly categorized crops into energy and protein plants, which is a differentiation that proved to be critical in the outcomes of our study. As stated before, the yield by crop together (as shown in Table 5) with the nutrient supply (as shown in Table 4) led to the nutrient supply by crop in kilogram per hectare (as shown in Table 6).

**Table 4. Relative nutrient supply by crops considered in linear optimization.**

| | | | | Protein and oil plants | | | | | | Energy plants | | | | | | |
|---|---|---|---|---|---|---|---|---|---|---|---|---|---|---|---|---|
| Nutrient yield with hydrolysis | | | | Soybean | Lupine | Field bean | Pea | Rapeseed | Sun | Maize | Wheat | Rye | Triticale | Barley | Oats | Potato |
| Glucose from starch (Legume) | | | 50% | | | | | | | | | | | | | |
| Glucose from starch (General) | | | 100% | | | | | | | | | | | | | |
| Amino acid yield (General) | | | 70% | | | | | | | | | | | | | |
| | | | Dry matter | 89.5% | 88.1% | 86.0% | 87.2% | 92.4% | 92.8% | 86.3% | 86.9% | 86.7% | 86.8% | 87.2% | 87.6% | 89.1% |
| | | | *Grams per 100 grams of fresh mass* | % | % | % | % | % | % | % | % | % | % | % | % | % |
| Nutrients | Saccharides | | Starch | 5.3 | 6.2 | 37.1 | 44.7 | 3.5 | 1.2 | 63.8 | 60 | 53.7 | 58.8 | 52.3 | 36.8 | 65.2 |
| | | | Sugar | 7.5 | 6 | 3.5 | 4.2 | 5.4 | 2.5 | 1.7 | 2.6 | 3.1 | 3 | 2.2 | 1.3 | 4.7 |
| | | | Starch + Sugar | 12.8 | 12.2 | 40.6 | 48.9 | 8.9 | 3.7 | 65.5 | 62.6 | 56.8 | 61.8 | 54.5 | 38.1 | 69.9 |
| | | | Glucose | 6.40 | 6.10 | 20.30 | 24.45 | 8.90 | 3.70 | 65.50 | 62.60 | 56.80 | 61.80 | 54.50 | 38.10 | 69.90 |
| | Amino acids | Essential | Arginine | 2.63 | 3.6 | 2.61 | 1.73 | 1.1 | 1.07 | 0.37 | 0.55 | 0.43 | 0.53 | 0.48 | 0.63 | 0.31 |
| | | | Cysteine | 0.53 | 0.55 | 0.34 | 0.28 | 0.47 | 0.26 | 0.19 | 0.27 | 0.2 | 0.26 | 0.23 | 0.31 | 0.12 |
| | | | Histidine | 0.98 | 0.73 | 0.68 | 0.51 | 0.47 | 0.37 | 0.22 | 0.25 | 0.18 | 0.24 | 0.23 | 0.2 | 0.14 |
| | | | Isoleucine | 1.68 | 1.55 | 1.09 | 0.85 | 0.77 | 0.59 | 0.28 | 0.39 | 0.3 | 0.38 | 0.36 | 0.36 | 0.28 |
| | | | Lysine | 2.24 | 1.63 | 1.72 | 1.48 | 1.15 | 0.6 | 0.23 | 0.32 | 0.34 | 0.39 | 0.37 | 0.4 | 0.41 |
| | | | Mehionin | 0.52 | 0.26 | 0.19 | 0.2 | 0.41 | 0.34 | 0.16 | 0.18 | 0.14 | 0.17 | 0.17 | 0.17 | 0.12 |
| | | | Phenylalanin | 1.82 | 1.3 | 1.12 | 0.96 | 0.72 | 0.63 | 0.37 | 0.52 | 0.38 | 0.43 | 0.48 | 0.47 | 0.33 |
| | | | Threonine | 1.46 | 1.24 | 0.95 | 0.78 | 0.89 | 0.57 | 0.29 | 0.33 | 0.29 | 0.33 | 0.34 | 0.33 | 0.32 |
| | | | Tryptophan | 0.46 | 0.23 | 0.22 | 0.18 | 0.24 | 0.2 | 0.05 | 0.13 | 0.08 | 0.13 | 0.12 | 0.12 | 0.08 |
| | | | Tyrosine | 1.28 | 1.56 | 0.85 | 0.63 | 0.53 | 0.37 | 0.32 | 0.34 | 0.22 | 0.31 | 0.28 | 0.34 | 0.35 |
| | | | Valine | 1.73 | 1.44 | 1.21 | 0.96 | 0.92 | 0.68 | 0.38 | 0.49 | 0.42 | 0.48 | 0.5 | 0.5 | 0.43 |
| | | Non-essential | Alanine | 1.53 | 1.13 | 1.1 | 0.9 | 0.84 | 0.65 | 0.56 | 0.4 | 0.38 | 0.42 | 0.41 | 0.45 | 0.32 |
| | | | Aspartic acid | 4.01 | 3.56 | 3.21 | 2.37 | 1.39 | 1.19 | 0.49 | 0.57 | 0.65 | 0.66 | 0.59 | 0.81 | 1.45 |
| | | | Glutamic acid | 6.34 | 6.94 | 4.72 | 3.32 | 3.07 | 2.46 | 1.41 | 3.15 | 1.92 | 2.4 | 2.24 | 1.61 | 1.38 |
| | | | Glycine | 1.55 | 1.33 | 1.15 | 0.9 | 0.92 | 0.85 | 0.29 | 0.45 | 0.38 | 0.44 | 0.4 | 0.48 | 0.28 |
| | | | Leucine | 2.72 | 2.43 | 2.03 | 1.45 | 1.28 | 0.87 | 0.93 | 0.74 | 0.52 | 0.64 | 0.68 | 0.69 | 0.46 |
| | | | Proline | 1.81 | 1.41 | 1.11 | 0.85 | 1.15 | 0.61 | 0.7 | 1.09 | 0.85 | 0.87 | 1.04 | 0.58 | 0.26 |
| | | | Serine | 1.89 | 1.79 | 1.36 | 0.95 | 0.81 | 0.64 | 0.38 | 0.54 | 0.38 | 0.45 | 0.41 | 0.46 | 0.32 |

## 3.4 Comparative energy demands

Energy requirements of CM production are compared with traditional meat, assessing the role of solar energy in reducing environmental impacts. Energy consumption forms a significant aspect of our study, given the high energy requirements of CM production, particularly the energy needed to maintain bioreactors at a consistent temperature of 37°C [39]. The energy

**Table 5. Yield by crops considered in linear optimization.**

| | | Protein and oil plants | | | | | | Energy plants | | | | | | |
|---|---|---|---|---|---|---|---|---|---|---|---|---|---|---|
| | | Soybean | Lupine | Field bean | Pea | Rapeseed | Sunflower | Maize | Wheat | Rye | Triticale | Barley | Oats | Potato |
| Average yield per hectare[a] | t FM/ha | 3.17 | 2.88 | 2.56 | 2.99 | 3.57 | 2.7 | 10.52 | 7.54 | 5.52 | 4.88 | 6.75 | 4.45 | 42.26 |
| Humidity of yields | % | 14% | 14% | 14% | 14% | 9% | 9% | 14% | 14% | 14% | 14% | 14% | 14% | 78% |
| Adjusted average yield per hectare | t FM/ha | 3.05 | 2.81 | 2.56 | 2.95 | 3.52 | 2.65 | 10.48 | 7.46 | 5.48 | 4.84 | 6.66 | 4.37 | 10.43 |

[a] LfL (2023) [82]

Note: t FM/ha = tons of fresh matter per hectare.

**Table 6. Nutrient supply by crops considered in linear optimization.**

| | | | | Protein and oil plants | | | | | | Energy plants | | | | | | | |
|---|---|---|---|---|---|---|---|---|---|---|---|---|---|---|---|---|---|
| **Nutrient yield with hydrolysis** | | | | Soybean | Lupine | Field bean | Pea | Rapeseed | Sunflower | Maize | Wheat | Rye | Triticale | Barley | Oats | Potato |
| Glucose from starch (Legume) | 50% | | | | | | | | | | | | | | | |
| Glucose from starch (General) | 100% | | | | | | | | | | | | | | | |
| Amino acid yield (General) | 70% | | | | | | | | | | | | | | | |
| | | | | kg/ha | kg/ha | kg/ha | kg/ha | kg/ha | kg/ha | kg/ha | kg/ha | kg/ha | kg/ha | kg/ha | kg/ha | kg/ha |
| Nutrients | Saccharides | | Glucose | 194.95 | 171.49 | 519.68 | 720.99 | 312.92 | 97.96 | 6866.65 | 4671.16 | 3110.05 | 2988.04 | 3628.13 | 1664.48 | 7293.76 |
| | Amino acids | Essential | Arginine | 56.08 | 70.85 | 46.77 | 35.71 | 27.07 | 19.83 | 27.15 | 28.73 | 16.48 | 17.94 | 22.37 | 19.27 | 22.64 |
| | | | Cysteine | 11.30 | 10.82 | 6.09 | 5.78 | 11.57 | 4.82 | 13.94 | 14.10 | 7.67 | 8.80 | 10.72 | 9.48 | 8.77 |
| | | | Histidine | 20.90 | 14.37 | 12.19 | 10.53 | 11.57 | 6.86 | 16.14 | 13.06 | 6.90 | 8.12 | 10.72 | 6.12 | 10.23 |
| | | | Isoleucine | 35.82 | 30.50 | 19.53 | 17.55 | 18.95 | 10.93 | 20.55 | 20.37 | 11.50 | 12.86 | 16.78 | 11.01 | 20.45 |
| | | | Lysine | 47.76 | 32.08 | 30.82 | 30.55 | 28.30 | 11.12 | 16.88 | 16.71 | 13.03 | 13.20 | 17.24 | 12.23 | 29.95 |
| | | | Methionin | 11.09 | 5.12 | 3.40 | 4.13 | 10.09 | 6.30 | 11.74 | 9.40 | 5.37 | 5.75 | 7.92 | 5.20 | 8.77 |
| | | | Phenylalanin | 38.81 | 25.58 | 20.07 | 19.82 | 17.72 | 11.68 | 27.15 | 27.16 | 14.56 | 14.55 | 22.37 | 14.37 | 24.10 |
| | | | Threonine | 31.13 | 24.40 | 17.02 | 16.10 | 21.90 | 10.56 | 21.28 | 17.24 | 11.12 | 11.17 | 15.84 | 10.09 | 23.37 |
| | | | Tryptophan | 9.81 | 4.53 | 3.94 | 3.72 | 5.91 | 3.71 | 3.67 | 6.79 | 3.07 | 4.40 | 5.59 | 3.67 | 5.84 |
| | | | Tyrosine | 27.29 | 30.70 | 15.23 | 13.00 | 13.04 | 6.86 | 23.48 | 17.76 | 8.43 | 10.49 | 13.05 | 10.40 | 25.56 |
| | | | Valine | 36.89 | 28.34 | 21.68 | 19.82 | 22.64 | 12.60 | 27.89 | 25.59 | 16.10 | 16.25 | 23.30 | 15.29 | 31.41 |
| | | Non-essential | Alanine | 32.62 | 22.24 | 19.71 | 18.58 | 20.67 | 12.05 | 41.10 | 20.89 | 14.56 | 14.21 | 19.11 | 13.76 | 23.37 |
| | | | Aspartic acid | 85.50 | 70.06 | 57.52 | 48.92 | 34.21 | 22.05 | 35.96 | 29.77 | 24.91 | 22.34 | 27.49 | 24.77 | 105.91 |
| | | | Glutamic acid | 135.18 | 136.58 | 84.58 | 68.53 | 75.56 | 45.59 | 103.47 | 164.54 | 73.59 | 81.23 | 104.38 | 49.24 | 100.80 |
| | | | Glycine | 33.05 | 26.17 | 20.61 | 18.58 | 22.64 | 15.75 | 21.28 | 23.51 | 14.56 | 14.89 | 18.64 | 14.68 | 20.45 |
| | | | Leucine | 58.00 | 47.82 | 36.38 | 29.93 | 31.50 | 16.12 | 68.25 | 38.65 | 19.93 | 21.66 | 31.69 | 21.10 | 33.60 |
| | | | Proline | 38.59 | 27.75 | 19.89 | 17.55 | 28.30 | 11.31 | 51.37 | 56.93 | 32.58 | 29.45 | 48.46 | 17.74 | 18.99 |
| | | | Serine | 40.30 | 35.23 | 24.37 | 19.61 | 19.94 | 11.86 | 27.89 | 28.21 | 14.56 | 15.23 | 19.11 | 14.07 | 23.37 |
| | | | Glutamine + Glutamine Acid | 135.18 | 136.58 | 84.58 | 68.53 | 75.56 | 45.59 | 103.47 | 164.54 | 73.59 | 81.23 | 104.38 | 49.24 | 100.80 |
| | | | Asparagine + Asparagine acid | 85.50 | 70.06 | 57.52 | 48.92 | 34.21 | 22.05 | 35.96 | 29.77 | 24.91 | 22.34 | 27.49 | 24.77 | 105.91 |
| | | | Sum of amino acids | 750.12 | 643.12 | 459.83 | 398.39 | 421.59 | 240.01 | 559.19 | 559.42 | 308.92 | 322.54 | 434.78 | 272.48 | 537.59 |
| | | | Sum of amino acids + glucose | 945.06 | 814.62 | 979.51 | 1119.38 | 734.51 | 337.97 | 7425.83 | 5230.58 | 3418.97 | 3310.59 | 4062.90 | 1936.96 | 7831.35 |

demand for traditional pork production is lower than that for CM production. To estimate the energy demands, we utilized data from several sources. For CM, an average value was obtained from three separate studies: 205.50 MJ/kg [40], 106.00 MJ/kg [50], and 48.00 MJ/kg [95], yielding an average of 119.83 MJ/kg. We averaged data from pork production LCAs by Mattick et al. (2015) [50], Dourmad et al. (2014) [97], Reckmann et al. [98], Nguyen et al. (2011) [99], and Wiedemann et al. (2018) [100], which led to a value of 17.16 MJ/kg for pork meat. This led to a differential of 102.67 MJ/kg comparing pork and CM production.

In light of the substantial energy differential between CM and pork production, we explored the prospect of integrating solar energy into the CM production process. Solar power was considered as the technology is mostly independent from location factors and most easy to scale, making it more flexible than other renewable energy sources like wind energy. Projected forecasts indicate the expansion of battery infrastructure in forthcoming years due to technological advancements and increased renewable energy integration [101, 102]. Recent trends have shown the performance of solar panels in the case study region of southern Germany to be approximately 1.35 ha MW/p [96] implying an estimated performance of 851,111.1 kWh/ha/yr for solar panels in this region. This energy yield was then implemented as part of the crop rotation within our linear programming model.

With this approach, we aim to navigate the energy challenge faced by CM production, presenting a holistic picture of the energy requirements and potential strategies for optimizing resource use in this emerging industry.

## 3.5 Scenarios

We present four CM production scenarios, each differing in crop extraction methods and solar energy integration to evaluate efficiency. Four scenarios were developed, each differing in the use of photovoltaic systems for energy supply and the extraction source for protein and sugar. Scenario 1 (max. extraction/no solar energy) involves the production of CM without the integration of photovoltaic systems. Instead, it relies solely on maximal extraction from crops for glucose and amino acids. This could be difficult to apply since it is challenging to fully extract both sugars and hydrolysates from crops. Scenario 2 (max. extraction/solar energy) enhances this approach by incorporating photovoltaic systems, harnessing solar energy to potentially reduce the carbon footprint and reliance on traditional energy sources, while maximizing crop extraction. Scenario 3 (specific extraction/no solar energy), in contrast, excludes photovoltaic systems and instead adopts a more targeted but less efficient extraction approach, utilizing protein-rich plants specifically as protein source and separate energy crops for the required glucose inputs, aiming for resource optimization. Scenario 4 (specific extraction/solar energy) combines the methodologies of Scenarios 2 and 3, employing photovoltaic systems for electricity production and distinctly utilizing protein plants for amino acids and energy crops for glucose sourcing.

The results are compared to several studies that conducted LCAs for pork production.

## 3.6 Consideration of carbon opportunity costs

The carbon opportunity cost of land use for CM is examined, incorporating forgone carbon sequestration into climate impact assessments. Food production and agriculture are responsible for a significant share, namely one-third, of all global greenhouse gas emissions, with the majority emanating from land use and land use change [103]. Carbon opportunity cost is an approach that integrates the forgone carbon sequestration potential of land allocated to specific uses like agriculture rather than allowing it to remain in or revert to a natural state. Every parcel of land has an intrinsic ability to capture and store carbon, particularly ecosystems like

forests, wetlands, or savannas. When these ecosystems are altered or converted, for instance into crop fields or pastures, the carbon that could have been sequestered by the native vegetation is a forgone benefit and therefore must be considered as an opportunity cost when quantifying the climate impact of a food product [104].

Searchinger et al. (2018) [68] suggest using the "carbon cost of consumption" (CCC) to evaluate the climate impact of food products. CCC is the sum of production emissions and carbon opportunity cost of a consumed product. Most previous studies only focus on production emissions. By factoring in carbon opportunity cost, we can quantify the climate impact of not only the production emissions from agriculture, but also the forgone benefit of native vegetation. This is especially relevant when comparing conventional animal agriculture with alternative food production methods like plant-based meat alternatives or CM, which potentially require less land and thereby incur lower carbon opportunity cost.

Incorporating carbon opportunity cost into climate impact assessments offers a more nuanced and comprehensive understanding of the climate impact of food production systems, particularly highlighting the value of native vegetation for climate change mitigation [68, 105]. The foregone carbon sequestrations potential of arable land in Germany is assumed with 2.9 t $CO_2$ ha$^{-1}$ yr$^{-1}$. Multiplying this factor with 44/12, which is the conversion factor from C to $CO_2$ [106], and the land requirements results in the carbon opportunity cost per output. The sum of total production emissions and carbon opportunity cost results in the carbon cost of consumption. Data for environmental impact of soy was taken from DonauSoja (2022) [107] and data of foregone carbon sequestration of arable land was taken from Fehrenbach & Bürck (2022) [108].

## 4. Results

The analysis of CM production, factoring in energy consumption and land use, yields compelling insights. We have found that solar energy integration and land utilization, coupled with crop rotation restrictions, play a critical role in the sustainability and feasibility of CM. In the ensuing results and discussion section, we elaborate on these findings, positioning them in the context of the existing body of research.

The crops' share in the 100-hectare reference land area following linear optimization is shown in Fig 1. In terms of crop distribution, all four scenarios utilized soybeans with a maximum cultivation area of 50 hectares. The cultivation of rapeseed was maximized within crop rotation limitations at 20 hectares for all scenarios. For wheat, scenario 1 had a cultivation area of 13.33 hectares, while scenario 2 had a cultivation area of 7.59 hectares. In contrast, scenarios 3 and 4 had a wheat cultivation area of 13.33 hectares and 8.75 hectares, respectively. For maize, scenario 1 had a cultivation area of 8.10 hectares and scenario 2 had a cultivation area of 9.63 hectares. In contrast, scenarios 3 and 4 have a cultivation area of 0 hectares for maize. Sunflower was cultivated at 13.33 hectares in scenarios 3 and 4 but was not cultivated in scenarios 1 and 2. Potato cultivation was 8.57 hectares for scenario 1, 7.04 hectares for scenario 2, and 3.33 hectares for scenarios 3 and 4. In terms of energy production, scenario 2 and 4 utilized solar panels with an area devoted to solar panels amounting to 5.74 hectares and 4.58 hectares, respectively.

A comparison of different scenarios for CM production and traditional pork farming was conducted, presented in Table 7. We made use of the review by Gislason et al. (2023) [21] to compare scenario outcomes to pork production worldwide. The review analyzed 31 LCAs calculating land use efficiency for pork production. Boxplots for land use efficiency reported in CM LCAs, Gislason et al. (2023) [21] and the scenarios of this study are visualized in Fig 2.

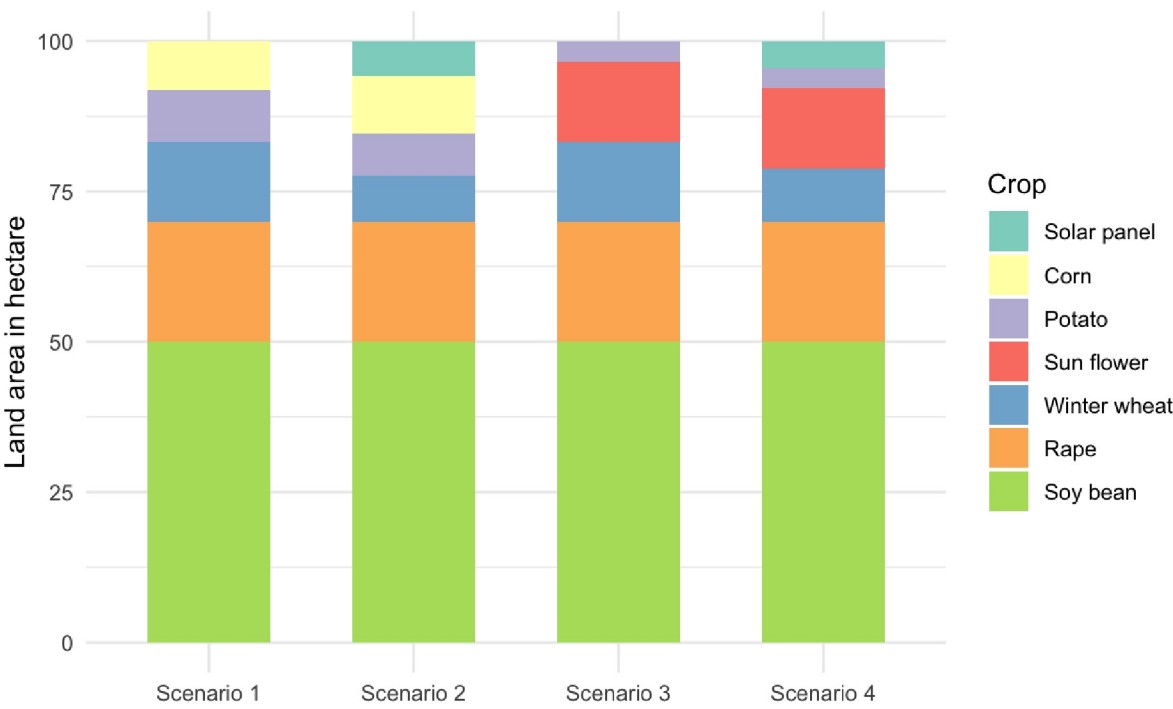

**Fig 1. Crop rotation for maximum cell media production in the four scenarios.**

The results showed differences in land use efficiency across the four scenarios. Scenario 1, which involved maximum CM production without solar panel energy and includes the extraction of all nutrients from the crops, had the highest efficiency of 1,785.16 kg/ha. Scenario 2, which added solar panels to the mix, showed a slightly lower efficiency of 1,727.14 kg/ha. Scenarios 3 and 4, which sourced nutrients only from respective protein and glucose plants, had efficiencies of 1,378.28 kg/ha and 1,378.28 kg/ha respectively. With scenarios 3 and 4, the production was inhibited by amino acid availability, more precisely by the lack of availability of lysine and alanine. Table 7 lists results of various LCAs to calculate a corrected area efficiency to fit various assumption of the observed LCAs. We used a carcass yield of 79% following Reckmann et al. (2013) [98].

On average, pork production also needs more land per kilogram than CM production looking at existing LCAs. Therefore, we can generally report slightly higher land use performance for CM which was also reported in aforementioned LCAs.

To calculate the carbon cost of consumption, which are the sum of production emissions and carbon opportunity cost [68], we used production emissions per area for the crops that are used to create the pig feed concentrates and the cell medium for each of the four scenarios. The results are shown in Table 8. Given the land use requirements for feed and cell mediums, we calculate the total production emissions per area. This result is combined with the output per area to derive the feed concentrate and cell medium production emissions. Regarding energy use, for pigs we use data based on Wirsenius et al. (2020) [105]. Manure management and enteric fermentation numbers for the pig farm are from Wirsenius et al. (2020) [105]. For CM, we assume no emissions in these two categories. All four mentioned categories of production emissions make up the total production emissions. Besides production emissions, we consider the carbon opportunity cost to quantify the carbon cost of consumption [68] for meat from pigs and the various CM scenarios.

**Table 7. Comparison of land use efficiency of different European pork production systems and cultivated meat production.**

| Scenario/ LCA | Protein source | Energy source | Solar panel use | Area efficiency (m²/kg) |
|---|---|---|---|---|
| Scenario 1 (max. extraction/ no solar energy) | Maximum yield from all crops | Maximum yield from crops | No | 5.6 |
| Scenario 2 (max. extraction/ solar energy) | Maximum yield from all crops | Maximum yield from crops | Yes | 5.79 |
| Scenario 3 (specific extraction/no solar energy) | Proteins only from protein crops | Sugar from energy crops | No | 7.26 |
| Scenario 4 (specific extraction/solar energy) | Proteins only from protein crops | Sugar from energy crops | Yes | 7.26 |
| Mean | | | | 6.48 |
| **Examples from LCAs for Cultivated Meat Production** | | | | |
| Smetana et al., 2015 | Cyanobacteria | Cyanobacteria | - | 0.4 |
| Vergeer et al., 2021 | Soy | Maize | - | 1.8 |
| Sinke et al., 2023 | 75% Soy hydrolysate/ 25% chemical & microbial production | Maize | - | 2.48 |
| Mattick, et al., 2015 | Soy | Maize | - | 5.4 |
| Tuomisto et al., 2022 | Soy | Maize | - | 6.89 |
| Mean (without ranges) | | | | 3.39 |
| **Average from LCAs for pork production** | | | | |
| Gislason et al., 2023 | 31 studies from various countries | | | 8.02 (adjusted from live weight) |
| Mean (without ranges) | | | | 7.93 |

Note: m²/kg = square meters per kilogram; LCA = Life Cycle Assessment.

The results indicate that when considering production emissions and carbon opportunity cost, scenarios 1 and 2 show slightly lower carbon cost of consumption (9.0 and 9.1 t $CO_2$e t$^{-1}$) as well as scenarios 3 and 4 (10.8 and 10.7 t $CO_2$e t$^{-1}$) compared to the emissions of German pork production (11.24 t $CO_2$e t$^{-1}$) reported by Wirsenius et al. (2020) [105].

## 5. Discussion

### 5.1 Comparison with existing literature

Our investigation into CM production in southern Germany reveals limited land use advantages when compared with traditional pork production when feedstock is solely taken from crops. Our findings contrast with the assertions of several previously published studies which have not taken into account crop rotation restrictions or pure plant-based cell media [40, 41, 50]. The examined scenarios highlight that crop rotation restrictions along with protein and sugar content of specific crops are crucial factors determining the land use efficiency of CM. If CM were fed by cell media produced from plant-based crops to a large extent, land use efficiency would not be substantially better than it would be for conventional pork production. It is important to keep in mind that the cell culture medium accounts for the highest environmental impact in CM production [42].

The higher energy demand of CM production translates into additional land requirements when renewable energy sources, namely ground-mounted photovoltaic systems, are employed.

## Boxplots of Different Pork Production Categories

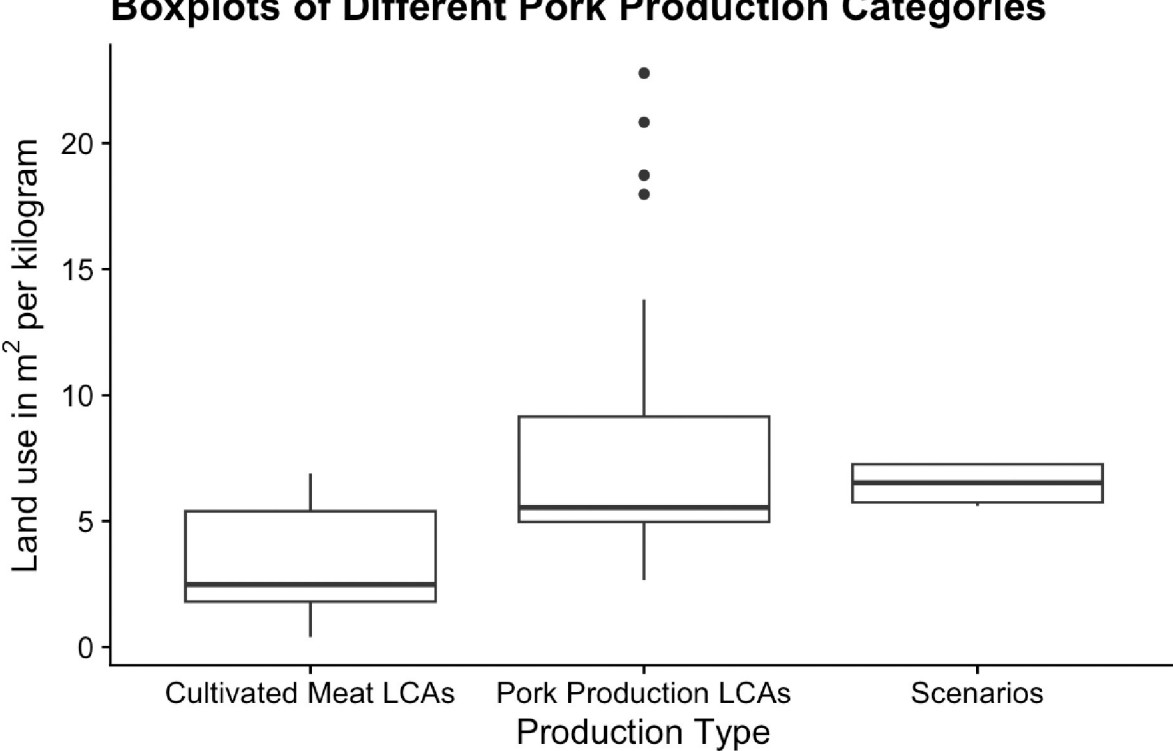

**Fig 2. Boxplots of land use efficiency for previous cultivated meat LCAs, pork production LCAs, and cultivated meat scenarios from this study.**

This is due to the additional land that solar panels would require. Since land use is a critical determinant of the climatic impact of pork (up to 75%) [105], substantial environmental benefits from CM in this aspect cannot be readily anticipated when agricultural feedstock is used. Also, for the carbon cost of consumption (production emissions and carbon opportunity cost) there is only a limited advantage of the considered CM scenarios in comparison to pig farming in Germany. In previous LCAs Sinke et al., 2023 brings up freed up land, which can only be realized if precision fermentation is employed as they use 25% for media production through biotechnology and 75% amounts of amino acids through soy hydrolysate which again on a global scope would not reflect a sustainable crop rotation [41]. As a consequence, public and private research needs to focus on CM production systems where proteins for cell media feedstock are produced through synthetic processes independent from agricultural products, like precision fermentation [61, 70].

Precision fermentation is essential for supplying nutrients to CM bioreactors, but scaling this technology presents major obstacles. Global fermentation capacity is currently inadequate to meet future demands, and engineering highly productive, robust microbial strains remains complex and costly [109]. Production costs are still too high, and significant regulatory hurdles persist, although they are somewhat less severe than those for CM [110].

Ensuring the purity of amino acids for bioreactor use is another challenge, particularly when crop-based substrates require extensive refinement, potentially leading to higher cost in precision fermentation [111, 112]. Additionally, fermentation and chemical processes are water-intensive, complicating sustainability efforts [41].

**Table 8. Land use, carbon opportunity cost, and carbon cost of consumption of 100 ha arable land for cultivated meat.**

| | | Scenario 1 (max. extraction/ no solar energy) | Scenario 2 (max. extraction/ solar energy) | Scenario 3 (specific extraction/ no solar energy) | Scenario 4 (specific extraction/ solar energy) | Unit | Crop yield t/ha | Crop production emissions t $CO_2$e $t^{-1}$ fw | Sources |
|---|---|---|---|---|---|---|---|---|---|
| Land use | Soybeans | 50.0 | 50.0 | 50.0 | 50.0 | ha | 3.2 | 0.4 | LFL, 2023; DonauSoja, 2022 |
| | Winter oilseed rapeseed | 20.0 | 20.0 | 20.0 | 20.0 | ha | 3.6 | 0.5 | LFL, 2023 |
| | Sunflower | - | - | 13.3 | 13.3 | ha | 2.7 | 0.8 | LFL, 2023 |
| | Maize for grain | 8.1 | 9.6 | - | - | ha | 10.5 | 0.3 | LFL, 2023 |
| | Winter wheat | 13.3 | 7.6 | 13.3 | 8.8 | ha | 7.5 | 0.3 | LFL, 2023 |
| | Potatoes | 8.6 | 7.0 | 3.3 | 3.3 | ha | 42.3 | 0.2 | LFL, 2023 |
| | Photovoltaic cells | - | 5.7 | - | 4.6 | ha | | | Böhm et al., 2022 |
| | **Total land use** | **100.0** | **100.0** | **100.0** | **100.0** | **ha** | | | |
| | Production emissions per area | 2.2 | 2.0 | 1.8 | 1.7 | t $CO_2$e $ha^{-1}$ | | | Own calculation |
| | Land requirement per output | 5.6 | 5.8 | 7.3 | 7.3 | $m^2$/kg | | | Own calculation |
| | Output per area | 1.8 | 1.7 | 1.4 | 1.4 | t/ha | | | Own calculation |
| Production emissions per output | Feed concentrates or medium production emissions | 1.2 | 1.2 | 1.3 | 1.2 | t $CO_2$e $t^{-1}$ | | | Own calculation |
| | Energy use, scaffold, equipment & other | 1.8 | 1.8 | 1.8 | 1.8 | t $CO_2$e $t^{-1}$ | | | Sinke et al., 2023 |
| | **Total production emissions** | **3.0** | **3.0** | **3.1** | **3.0** | t $CO_2$e $t^{-1}$ | | | Own calculation |
| Carbon Opp. Cost | Foregone carbon sequestration of arable land | 2.9 | 2.9 | 2.9 | 2.9 | t C/ ha*a | | | Fehrenbach & Bürck, 2022 |
| | Foregone carbon sequestration of arable land | 10.6 | 10.6 | 10.6 | 10.6 | t $CO_2$/ ha*a | | | Own calculation |
| | **Carbon opportunity cost** | **6.0** | **6.2** | **7.7** | **7.7** | t $CO_2$e $t^{-1}$ | | | Own calculation |
| | **Carbon cost of consumption** | **9.0** | **9.1** | **10.8** | **10.7** | t $CO_2$e $t^{-1}$ | | | Own calculation |

## 5.2 Limitations

While our study provides valuable insights into the land use implications of CM, it is important to note a few limitations. Our case study analysis is based on southern Germany, so the results may not be representative of other geographical locations with different agricultural systems, climates, and land accessibility. More agricultural crops could be grown and used as media bases, even in a German context. However, every agricultural site offers different basic requirements. We limited our model to a set of regular crops used in Germany.

Furthermore, our data was based on German yield averages. Future integrated and regenerative agricultural practices could lead to different results, although lower crop yields and therefore lower sugar and amino acid yields are to be expected with these cultivation methods [113]. Moreover, conservation agriculture does not generally lead to $N_2O$ emission mitigation [114].

The economic feasibility of CM production is heavily influenced by the cost of growth media, which remains a major financial bottleneck. Current estimates indicate that media expenses account for up to 99% of material costs, driven by expensive amino acids and growth factors produced through precision fermentation or chemical synthesis [37, 41, 115]. Even as bioreactor scale increases, reducing media costs is critical to achieving competitive prices with conventional meat [116]. While precision fermentation has lower land use impacts, its complexity and high production costs makes it still a challenge. Chemical production of growth factors adds to this economic burden. Emerging research suggests that plant-based media could offer a more cost-effective solution, though this remains speculative as hydrolysates have not fully proven feasibility for cell media formulation [37, 115].

The study assumes direct extraction of amino acids and glucose from crops, which oversimplifies real-world processes and might overestimate yields, thus affecting production efficiency and land use metric. Additionally, incorporating biotechnological methods for amino acid production could realistically enhance efficiency. Other studies using precision fermentation technologies showed more optimistic scenarios that could be implemented in the future. Those hybrid media formulations could decrease land use and efficiency [41]. On the other hand, involving synthetic amino acid supply for pig feed would also boost efficiency of land use for pork production [117, 118].

In general, we did not consider the upcycling of by-products from food processing, such as soybean meal, rapeseed meal, or sugar beet pulp. There may still be potential to increase land use efficiency in the production of plant-based cell media. Many co-products are created by industrial food processing leading to sugar and protein rich feed that could be used for cultivated as they are used for animal feed now. But many sustainability criteria must be reconsidered for those side streams anew [119]. While land use and emissions could be decreased, economic feasibility often remains a constraint when it comes to adopting side stream products and their usage. Currently available solutions include alternatives to fermentable sugar, nitrogen, and phosphates, which could also be converted into proteins through precision fermentation [120]. During the production of hydrolysates and glucose from crops, byproducts are generated including solid residues (stillage), fiber, proteins, lipids, ash, and in the case of maize, maize gluten meal, maize gluten feed, and maize oil. One limitation of our study lies in the fact that we did not account for the metabolites [50] produced by cells during the CM production process. These metabolites, primarily consisting of compounds such as urea, ammonia, lactic acid, and other organic acids [121], could potentially be utilized as a form of manure or soil conditioner. This factor was not incorporated into our current model and could represent an area of interest for further investigation.

In terms of energy, CM comes with higher energy needs than animal meats, including pork [39]. We only considered land use scenarios if solar energy was used, leading to direct influence on land use. While it might use less land per output, the use of fossil energy for production of CM would have a far greater impact on the environment [122]. Furthermore, the model considers only land use and does not account for other critical environmental impacts, such as water usage, influence on biodiversity, or ecosystem structures.

## 5.3 Future research directions

There are several opportunities for future research on agricultural involvement in CM production. Agriculture will be involved in CM production in any matter. It is also critical to understand the social acceptance and economic viability of CM. Thus, future studies could focus on consumer perceptions, market dynamics, and the broader socio-economic implications of a shift from conventional meat production to CM.

Future research could build upon the findings of this study by examining the environmental impacts of CM in a more comprehensive and holistic manner, such as by incorporating a range of sustainability metrics beyond land use and energy consumption. To examine CMs full sustainability impacts this will be crucial. This could include additional factors, such as water use, biodiversity impacts, and further potential for carbon sequestration.

Given the constraints discussed above regarding information on extraction efficiencies, research is needed to provide more precise estimates of nutrient extraction efficiencies. This would enhance the accuracy and relevance of the model and therefore the results. Taken together, these research directions would contribute to a more comprehensive understanding of the potential of CM as a sustainable alternative to conventional meat production and provide valuable insights for policy makers, industry stakeholders, and consumers alike.

## 6. Conclusions

This study evaluates the potential role of agriculture to produce the feedstock for CM production. Using a linear programming model, we investigated different scenarios of CM production based on agricultural feedstock in southern Germany, considering crop rotation restrictions and energy requirements, particularly the integration of solar energy.

Our findings indicate that CM production using crops as raw materials for cell media does not offer significant land use efficiency improvements over conventional pork production. Scenarios with maximum nutrient extraction from crops showed only minor land use benefits, while scenarios relying on specific crops for glucose and amino acids were less efficient. Additionally, the higher energy demands of CM production, especially when integrating solar energy, translate into further land requirements, thus not significantly advancing climate change mitigation.

This study suggests that agriculture's role in directly providing protein feedstock for CM production should be small to ensure sustainability of CM. Policymakers and industry stakeholders should focus on exploring advanced biotechnological methods i.e. precision fermentation for cell media production to maximize the efficiency of CM. Further research should investigate systems where protein sources are based on precision fermentation rather than directly from crops.

## Author Contributions

**Conceptualization:** Peter Breunig.

**Data curation:** Hanno Kossmann, Thorsten Moess.

**Formal analysis:** Hanno Kossmann, Thorsten Moess.

**Methodology:** Hanno Kossmann.

**Visualization:** Hanno Kossmann.

**Writing – original draft:** Hanno Kossmann.

**Writing – review & editing:** Hanno Kossmann, Peter Breunig.

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
