## [Decision Letter · Decision Letter 0]

23 Oct 2024

PONE-D-24-43823The Climate Impact and Land Use of Cultivated Meat: Evaluating Agricultural Feedstock ProductionPLOS ONE

Dear Dr. Kossmann,

Thank you for submitting your manuscript to PLOS ONE. After careful consideration, we feel that it has merit but does not fully meet PLOS ONE’s publication criteria as it currently stands. Therefore, we invite you to submit a revised version of the manuscript that addresses the points raised during the review process.

We look forward to receiving your revised manuscript.

Kind regards,

António Raposo

Academic Editor

PLOS ONE

Journal Requirements:

1. When submitting your revision, we need you to address these additional requirements. Please ensure that your manuscript meets PLOS ONE's style requirements, including those for file naming. The PLOS ONE style templates can be found at https://journals.plos.org/plosone/s/file?id=wjVg/PLOSOne_formatting_sample_main_body.pdf and https://journals.plos.org/plosone/s/file?id=ba62/PLOSOne_formatting_sample_title_authors_affiliations.pdf 2. Thank you for stating the following in the Acknowledgments Section of your manuscript: "Funding for research was provided by the Bavarian State Government as part of the High-Tech Agenda Bavaria." We note that you have provided funding information that is not currently declared in your Funding Statement. However, funding information should not appear in the Acknowledgments section or other areas of your manuscript. We will only publish funding information present in the Funding Statement section of the online submission form. Please remove any funding-related text from the manuscript and let us know how you would like to update your Funding Statement. Currently, your Funding Statement reads as follows: “The authors received no specific funding for this work.” Please include your amended statements within your cover letter; we will change the online submission form on your behalf.

Additional Editor Comments:

Dear authors, please revise your manuscript according to both reviewers' suggestions.

Reviewers' comments:

Reviewer's Responses to Questions

**Comments to the Author**

1. Is the manuscript technically sound, and do the data support the conclusions?

Reviewer #1: Partly

Reviewer #2: Yes

2. Has the statistical analysis been performed appropriately and rigorously? 

Reviewer #1: Yes

Reviewer #2: Yes

3. Have the authors made all data underlying the findings in their manuscript fully available?

Reviewer #1: Yes

Reviewer #2: Yes

4. Is the manuscript presented in an intelligible fashion and written in standard English?

Reviewer #1: Yes

Reviewer #2: Yes

5. Review Comments to the Author

Reviewer #1: Comments

The manuscript addresses a valid research question and is within the scope of the journal. In addition, this article addresses a pertinent question concerning the evaluation of cultivated meat production based on agricultural raw material and its economic and environmental impacts.

The title clearly indicates the focus of the manuscript and is concise. The abstract is precise in terms of its objectives, methods, results and conclusion.

To enhance the development and structure of the text, I'd like to provide you with some comments, questions and suggestions.

1. In the introductions of each section, such as the Background of CM and Materials and Methods, I suggest beginning by clearly identifying what will be described. Below, I have provided an example of how this can be adjusted as a suggestion.

- In the Background of CM: "This section outlines" could be changed to "Technical and environmental aspects of MC production, innovative approaches, regulatory challenges, and the importance of transparent and accurate information for consumer acceptance will be described."

- In Methods and Scenarios, I suggest starting directly with the description of items 3.1, 3.2, and 3.3 without an introductory text, as each subsection will play the role of explaining and contextualizing the content.

2 In Table 1:

- Clarify in the caption what (t/ha) stands for.

3. In line 412 and line 421:

- I suggest removing the word "see" from the expressions (see Table 2) and (see Table 3), as it is unnecessary; simply (Table 2) and (Table 3) are sufficient.

4. Clarify what the asterisk (*) means, as referenced in line 440.

5. In Table 3:

- Explain what "wm" refers to in "g/g CM wm stoich. Demand" in the caption.

6. Provide an explanation or reference for what the "INRAE-CIRAD-AFZ" table refers to.

7. In Table 4:

- I suggest formatting the text using the same font color; if you'd like to highlight something, use bold.

- Additionally, the borders should be adjusted to match in color, line thickness, and formatting, consistent with the other tables.

- All text should start with an uppercase letter.

- Include in the caption what DM stands for.

- If words are underlined, I recommend standardizing them. If you want to provide information on these underlined items, include them in plain text.

8. In lines 478 to 481:

- There are duplicate texts; I suggest removing them.

9. In Table 5:

- Standardize the words within parentheses by capitalizing the first letter in all lines.

- Explain these units of measurement in the caption: t FM/ha.

- I suggest removing "corr. Ø" and explaining these items clearly.

10. In Table 6: Follow the same standardization as the previous tables.

- I suggest formatting the text using the same font color; if you'd like to highlight something, use bold.

- Additionally, the borders should be adjusted to match in color, line thickness, and formatting, consistent with the other tables.

- All text should start with an uppercase letter.

- If words are underlined, I recommend standardizing them. If you want to provide information on these underlined items, include them in plain text.

11. In Table 7:

- Clarify in the caption what LCA stands for.

12. I suggest adding a separate section on limitations and future perspectives, which would bring clarity and flow to the text.

13. I recommend discussing a few important points, such as:

- In addition to production and environmental costs, where would the financial costs fit in? Where would the funding come from?

- In this study, was there specific funding for obtaining the raw materials? If so, I suggest describing it.

- For more extensive treatments in this production, how would the product’s purity level be achieved?

- Regarding the practical viability of large-scale cultured meat production, including considerations of cost and resource demand, I suggest discussing these aspects.

14. In Acknowledgements:

- You mention that funding for research was provided by the Bavarian State Government as part of the Tech Agenda Bavaria.

- What specifically was this funding for? I suggest describing this in the Methods section and also updating the initial checklist where it is declared that no funding was received.

Reviewer #2: The study made by Kossmann and colleagues presents a critical evaluation of cultivated meat (CM) production based on agricultural feedstock and offers valuable insights into its limitations and future directions. The primary objective—evaluating land use efficiency—is highly relevant, especially as sustainability concerns rise with the increasing global demand for meat.

One of the main findings is that using crops as the raw material for cell media does not significantly improve land use efficiency or climate change mitigation compared to conventional pork production. This may indicate that the initial assumption—using crops for cell media—is inherently limited in its sustainability. Future research should not only consider but prioritize alternatives like precision fermentation from the start.

While the study acknowledges the role of solar energy and the high energy demands of CM production, it seems to underemphasize the broader energy costs involved in cultivating meat. The focus on land use efficiency might obscure the significant environmental footprint associated with energy consumption in these production systems, especially as the energy sources transition from fossil fuels to renewables.

The study rightly considers crop rotation, but it doesn't explore more innovative or regenerative agricultural practices that could reduce land requirements or enhance sustainability. This should be considered in the discussion.

There is no discussion of the economic feasibility of CM production using agricultural feedstock. While the study focuses on land use, understanding the real financial costs would be essential for assessing the viability of these studies.

The study touches on climate change mitigation but does not provide a detailed analysis of other environmental factors such as water use, biodiversity, or soil health impacts from agricultural feedstock use. CM's sustainability should be evaluated through a broader lens of environmental indicators.

While the study recommends shifting towards precision fermentation, it doesn't provide enough detail on the challenges or opportunities of adopting this technology. A more nuanced exploration of how precision fermentation and other biotechnological advancements can improve the sustainability of CM and what barriers (technical, regulatory, economic) need to be overcome would make the study more actionable.

6. PLOS authors have the option to publish the peer review history of their article (what does this mean?). If published, this will include your full peer review and any attached files.

Reviewer #1: **Yes: **Marcela Gomes Reis

Reviewer #2: **Yes: **M. João Reis Lima

---

## [Author Response · Author response to Decision Letter 0]

25 Nov 2024

PLOS ONE

The Climate Impact and Land Use of Cultivated Meat: Evaluating Agricultural Feedstock Production - PONE-D-24-43823

Reviewers‘ Comments

Reviewer 1 

In the introductions of each section, such as the Background of CM and Materials and Methods, I suggest beginning by clearly identifying what will be described. Below, I have provided an example of how this can be adjusted as a suggestion.

- In the Background of CM: "This section outlines" could be changed to "Technical and environmental aspects of MC production, innovative approaches, regulatory challenges, and the importance of transparent and accurate information for consumer acceptance will be described."

- In Methods and Scenarios, I suggest starting directly with the description of items 3.1, 3.2, and 3.3 without an introductory text, as each subsection will play the role of explaining and contextualizing the content. 

Thank you for your valuable feedback. We have revised the introductions for each section, as suggested, to clearly identify the content that follows. In the Background of CM, we now specify the technical and environmental aspects, innovative approaches, regulatory challenges, and the importance of transparency in consumer acceptance. For Methods and Scenarios, we incorporated brief, direct introductions in each subsection (3.1, 3.2, and 3.3) to effectively contextualize the content without additional introductory text. We believe these adjustments enhance the clarity and flow as recommended.

In Table 1:

- Clarify in the caption what (t/ha) stands for. We have revised the caption for Table 1 to clarify that yields are presented in tons per hectare (t/ha) as suggested. The updated caption now reads: "Average crop yields (in tons per hectare, t/ha) in southern Germany, Germany, the world, and continents in 2021." This addition should improve clarity for readers. 16

In line 412 and line 421:

- I suggest removing the word "see" from the expressions (see Table 2) and (see Table 3), as it is unnecessary; simply (Table 2) and (Table 3) are sufficient. Thank you for the suggestion. We have removed the word "see" from the expressions in lines 412 and 421, so they now read simply as "(Table 2)" and "(Table 3)" to improve conciseness.

Clarify what the asterisk (*) means, as referenced in line 440.

Thank you for pointing this out. We have replaced the asterisk (*) with a multiplication symbol (×) in line 440 to enhance clarity. This adjustment should make the notation more straightforward for readers.

In Table 3:

- Explain what "wm" refers to in "g/g CM wm stoich. Demand" in the caption. 

We have updated the caption for Table 3 to clarify the abbreviations. It now reads: "Relative nutrient requirements for cell mass production g/g CM = gram per gram cultivated meat; FM = Fresh mass)." This should improve clarity for readers.

Provide an explanation or reference for what the "INRAE-CIRAD-AFZ" table refers to.

Thank you for the suggestion. We have added an explanation for the INRAE-CIRAD-AFZ Feed Tables, clarifying its purpose and origin. Additionally, we noticed a duplication of the sentence below Table 3 and have removed it for conciseness.

In Table 4:

- I suggest formatting the text using the same font color; if you'd like to highlight something, use bold.

- Additionally, the borders should be adjusted to match in color, line thickness, and formatting, consistent with the other tables.

- All text should start with an uppercase letter.

- Include in the caption what DM stands for.

- If words are underlined, I recommend standardizing them. If you want to provide information on these underlined items, include them in plain text. 

Thank you for the detailed feedback. We have implemented the suggested changes in Table 4 by standardizing to black font color with bold for emphasis, adjusting border color, line thickness, and formatting to match the other tables. We have also ensured all text begins with uppercase letters, spelled out “DM”, and removed any underlining for consistency. These adjustments should enhance readability and uniformity across the tables.

In lines 478 to 481:

- There are duplicate texts; I suggest removing them. Thank you for catching that oversight. We have removed the duplicate text in lines 478 to 481 as suggested. 22

In Table 5:

- Standardize the words within parentheses by capitalizing the first letter in all lines.

- Explain these units of measurement in the caption: t FM/ha.

- I suggest removing "corr. Ø" and explaining these items clearly. Thank you for the helpful comments. We have standardized the capitalization within parentheses in Table 5. Additionally, we clarified the units of measurement in the caption, explaining "t FM/ha" as tons per hectare of fresh matter, and rephrased to remove "corr. Ø," providing a clear explanation of adjusted average yields.

In Table 6: Follow the same standardization as the previous tables.

- I suggest formatting the text using the same font color; if you'd like to highlight something, use bold.

- Additionally, the borders should be adjusted to match in color, line thickness, and formatting, consistent with the other tables.

- All text should start with an uppercase letter.

- If words are underlined, I recommend standardizing them. If you want to provide information on these underlined items, include them in plain text.

Thank you for your feedback. We have applied the same standardization as in previous tables by using a single font color with bold for emphasis, adjusting borders for consistency, capitalizing all text at the start, and removing underlining.

In Table 7:

- Clarify in the caption what LCA stands for. Thank you for the suggestion. We have clarified in the caption that "LCA" stands for Life Cycle Assessment to improve understanding for readers.

I suggest adding a separate section on limitations and future perspectives, which would bring clarity and flow to the text.

 We separated the “Discussion” section into three parts to improve clarity and flow.

I recommend discussing a few important points, such as:

- In addition to production and environmental costs, where would the financial costs fit in? Where would the funding come from?

- In this study, was there specific funding for obtaining the raw materials? If so, I suggest describing it.

- For more extensive treatments in this production, how would the product’s purity level be achieved?

- Regarding the practical viability of large-scale cultured meat production, including considerations of cost and resource demand, I suggest discussing these aspects.

Thank you for your valuable feedback. We have addressed your points as follows:

• Financial Costs and Funding Sources: We have added a discussion on the economic aspects of cultivated meat (CM) production, highlighting that the high cost of growth media is a significant financial barrier. This includes the challenges of funding and the need for cost-effective solutions to make CM competitive with conventional meat. Details are included in the revised “2.3 Economic viability and impacts on the meat industry and farming communities” section and in the “5.2 Limitations” section.

• Funding for Raw Materials: Our study is a theoretical modeling analysis that utilized existing data, so there was no need for specific funding to obtain raw materials. We have clarified this in the "Methods and Scenarios" section and expanded the "Acknowledgements" to describe the funding that supported our research.

• Achieving Product Purity: We have included a discussion on the challenges of achieving the necessary purity levels for amino acids and other nutrients in large-scale CM production. This addresses the technical and cost implications of refining crop-based substrates to meet bioreactor requirements, as detailed in the "5.1 Comparison with existing literature" section.

In Acknowledgements:

- You mention that funding for research was provided by the Bavarian State Government as part of the Tech Agenda Bavaria.

- What specifically was this funding for? I suggest describing this in the Methods section and also updating the initial checklist where it is declared that no funding was received.

We added a clearer explanation to the funding obtained: 

“Funding for the author's research position, which supported the completion of this study, was provided by the Bavarian State Government as part of the High-Tech Agenda Bavaria.”

Reviewer 2 

While the study acknowledges the role of solar energy and the high energy demands of CM production, it seems to underemphasize the broader energy costs involved in cultivating meat. The focus on land use efficiency might obscure the significant environmental footprint associated with energy consumption in these production systems, especially as the energy sources transition from fossil fuels to renewables. Thank you for highlighting the need to address the broader energy costs in cultivated meat (CM) production. We agree that focusing solely on land use efficiency may overlook the significant environmental footprint and cost associated with the high energy demands of CM, especially during the transition from fossil fuels to renewable energy sources.

To address this, we have added a brief discussion in the manuscript under the “Limitations” section 5.2.

The study rightly considers crop rotation, but it doesn't explore more innovative or regenerative agricultural practices that could reduce land requirements or enhance sustainability. This should be considered in the discussion. 

Thank you for your valuable suggestion regarding the exploration of innovative and regenerative agricultural practices. We agree that considering these practices could provide additional insights into reducing land requirements and enhancing sustainability in cultivated meat (CM) production.

To address this, we have added a discussion in the manuscript under the "5.2 Limitations” section:

"Furthermore, our data was based on German yield averages. Future integrated and regenerative agricultural practices could lead to different results, although lower crop yields and therefore lower sugar and amino acid yields are to be expected with these cultivation methods [113]. Moreover, conservation agriculture does not generally lead to N₂O emission mitigation [114]."

There is no discussion of the economic feasibility of CM production using agricultural feedstock. While the study focuses on land use, understanding the real financial costs would be essential for assessing the viability of these studies.

We addressed your comment, in the newly added “5.2 Limitations” section. In this section, we discuss the significant financial challenges associated with CM production:

• High Cost of Growth Media: We highlight that the cost of growth media remains a major financial bottleneck in CM production. Current estimates indicate that media expenses can account for up to 99% of material costs, primarily due to the expensive amino acids and growth factors produced through precision fermentation or chemical synthesis [37,41,115].

• Challenges with Precision Fermentation: While precision fermentation can reduce land use impacts, its complexity and high production costs pose significant challenges. The production of growth factors through chemical synthesis adds to the economic burden [37,115].

• Potential of Plant-Based Media: Emerging research suggests that plant-based media could offer a more cost-effective solution. However, this remains speculative as the feasibility of using hydrolysates for cell media formulation has not been fully established [37,115].

• Scaling Bioreactors: Even as bioreactor scales increase, reducing media costs is critical to achieving competitive prices with conventional meat [116].

The study touches on climate change mitigation but does not provide a detailed analysis of other environmental factors such as water use, biodiversity, or soil health impacts from agricultural feedstock use. CM's sustainability should be evaluated through a broader lens of environmental indicators.

Thank you for pointing out the need to evaluate cultivated meat (CM) sustainability through a broader range of environmental indicators beyond land use. We agree that factors such as water use, biodiversity, and soil health are essential components of a comprehensive assessment.

We addressed this in the end of the “5.2 Limitations” section: “Furthermore, the model considers only land use and does not account for other critical environmental impacts, such as water usage, influence on biodiversity, or ecosystem structures.”

And in the “5.3 Future research directions” section:

" Future research could build upon the findings of this study by examining the environmental impacts of CM in a more comprehensive and holistic manner, such as by incorporating a range of sustainability metrics beyond land use and energy consumption. To examine CMs full sustainability impacts this will be crucial. This could include additional factors, such as water use, biodiversity impacts, and further potential for carbon sequestration."

While the study recommends shifting towards precision fermentation, it doesn't provide enough detail on the challenges or opportunities of adopting this technology. A more nuanced exploration of how precision fermentation and other biotechnological advancements can improve the sustainability of CM and what barriers (technical, regulatory, economic) need to be overcome would make the study more actionable. 

Thank you. To address your comment, we have expanded the "5.1 Comparison with existing literature" section to include a detailed analysis of the challenges and opportunities associated with adopting precision fermentation. 

• Technical Challenges: We highlight the complexities in scaling up precision fermentation technologies, including inadequate global fermentation capacity and the difficulties in engineering productive and robust microbial strains [109].

• Economic Barriers: We address the high production costs, noting that media expenses can account for up to 99% of material costs due to expensive amino acids and growth factors produced through precision fermentation or chemical synthesis [37,41,115].

• Regulatory Hurdles: We discuss the significant regulatory challenges that persist, although they are somewhat less severe than those for CM itself [110].

• Environmental Considerations: We acknowledge that fermentation and chemical processes are water-intensive, which complicates sustainability efforts [41].

---

## [Decision Letter · Decision Letter 1]

4 Dec 2024

PONE-D-24-43823R1The Climate Impact and Land Use of Cultivated Meat: Evaluating Agricultural Feedstock ProductionPLOS ONE

Dear Dr. Kossmann,

Thank you for submitting your manuscript to PLOS ONE. After careful consideration, we feel that it has merit but does not fully meet PLOS ONE’s publication criteria as it currently stands. Therefore, we invite you to submit a revised version of the manuscript that addresses the points raised during the review process.

We look forward to receiving your revised manuscript.

Kind regards,

António Raposo

Academic Editor

PLOS ONE

Journal Requirements:

**Additional Editor Comments:**

Dear authors, I encourage you to revise your manuscript according to the reviewer 1 comments.

Reviewers' comments:

Reviewer's Responses to Questions

**Comments to the Author**

1. If the authors have adequately addressed your comments raised in a previous round of review and you feel that this manuscript is now acceptable for publication, you may indicate that here to bypass the “Comments to the Author” section, enter your conflict of interest statement in the “Confidential to Editor” section, and submit your "Accept" recommendation.

Reviewer #1: (No Response)

Reviewer #2: All comments have been addressed

2. Is the manuscript technically sound, and do the data support the conclusions?

Reviewer #1: Yes

Reviewer #2: Yes

3. Has the statistical analysis been performed appropriately and rigorously? 

Reviewer #1: Yes

Reviewer #2: Yes

4. Have the authors made all data underlying the findings in their manuscript fully available?

Reviewer #1: Yes

Reviewer #2: Yes

5. Is the manuscript presented in an intelligible fashion and written in standard English?

Reviewer #1: Yes

Reviewer #2: Yes

6. Review Comments to the Author

Reviewer #1: Most of the issues raised have been addressed; however, I suggest some revisions regarding the tables:

Table 1. It is acceptable to use "t/ha" alone, but I recommend adding the note: "t/ha: in tons per hectare" below the description of sources (a) and (b). The caption should be placed below the table, not in its title.

Table 3. Similarly, in Table 3, these notes should be added below source (a), described under the table, rather than in the title.

Table 4. The table still has borders that differ from the others; I suggest adjusting them.

Table 5. As with Tables 1 and 3, the notes I suggested for the caption should be added below source (a), described under the table, and not in the title.

Table 7. As with Tables 1, 3, and 5, the notes I suggested for the caption should be added below the table, not in its title.

Reviewer #2: After the reviews made by the authors, I believe that the manuscript can be published since the majority of the pointed concerns were answered and modified by the authors.

7. PLOS authors have the option to publish the peer review history of their article (what does this mean?). If published, this will include your full peer review and any attached files.

Reviewer #1: **Yes: **Marcela Gomes Reis

Reviewer #2: **Yes: **M. João Reis Lima

---

## [Author Response · Author response to Decision Letter 1]

5 Dec 2024

Thank you for your review and valuable suggestions regarding our tables. 

We have made the following revisions accordingly:

• Tables 1, 3, 5, and 7: We have moved the notes (e.g., “t/ha: tons per hectare”) below the description of sources placing them under the table rather than in the title.

• Tables 4, 5, and 6: We have adjusted the borders of Tables 4, 5, and 6 to ensure consistency with the formatting of the other tables.

---

## [Editor Report · Decision Letter 2]

8 Dec 2024

PONE-D-24-43823R2The Climate Impact and Land Use of Cultivated Meat: Evaluating Agricultural Feedstock ProductionPLOS ONE

Dear Dr. Kossmann,

Thank you for submitting your manuscript to PLOS ONE. After careful consideration, we feel that it has merit but does not fully meet PLOS ONE’s publication criteria as it currently stands. Therefore, we invite you to submit a revised version of the manuscript that addresses the points raised during the review process.

We look forward to receiving your revised manuscript.

Kind regards,

António Raposo

Academic Editor

PLOS ONE

Journal Requirements:

Additional Editor Comments:

Dear authors,

I encourage you to address the comments provided by reviewer 1 and resubmit the revised manuscript. Thank you!

Reviewer 1:

Most of the issues raised have been addressed; however, I suggest some revisions regarding the tables:

Table 1. It is acceptable to use "t/ha" alone, but I recommend adding the note: "t/ha: in tons per hectare" below the description of sources (a) and (b). The caption should be placed below the table, not in its title.

Table 3. Similarly, in Table 3, these notes should be added below source (a), described under the table, rather than in the title.

Table 4. The table still has borders that differ from the others; I suggest adjusting them.

Table 5. As with Tables 1 and 3, the notes I suggested for the caption should be added below source (a), described under the table, and not in the title.

Table 7. As with Tables 1, 3, and 5, the notes I suggested for the caption should be added below the table, not in its title.

---

## [Author Response · Author response to Decision Letter 2]

10 Dec 2024

Thank you for your review and valuable suggestions regarding our tables. 

We have made the revisions accordingly. We incorporated your suggestions as outlined. If there is anything you feel we have not fully addressed, could you please clarify what specific changes are still needed?

We added “Note: m²/kg = square meters per kilogram; LCA = Life Cycle Assessment.” to Table 7.

---

## [Decision Letter · Decision Letter 3]

12 Dec 2024

The Climate Impact and Land Use of Cultivated Meat: Evaluating Agricultural Feedstock Production

PONE-D-24-43823R3

Dear Dr. Kossmann,

We’re pleased to inform you that your manuscript has been judged scientifically suitable for publication and will be formally accepted for publication once it meets all outstanding technical requirements.

Kind regards,

António Raposo

Academic Editor

PLOS ONE

Additional Editor Comments (optional):

Reviewers' comments:

Reviewer's Responses to Questions

**Comments to the Author**

1. If the authors have adequately addressed your comments raised in a previous round of review and you feel that this manuscript is now acceptable for publication, you may indicate that here to bypass the “Comments to the Author” section, enter your conflict of interest statement in the “Confidential to Editor” section, and submit your "Accept" recommendation.

Reviewer #1: All comments have been addressed

2. Is the manuscript technically sound, and do the data support the conclusions?

Reviewer #1: Yes

3. Has the statistical analysis been performed appropriately and rigorously? 

Reviewer #1: Yes

4. Have the authors made all data underlying the findings in their manuscript fully available?

Reviewer #1: Yes

5. Is the manuscript presented in an intelligible fashion and written in standard English?

Reviewer #1: Yes

6. Review Comments to the Author

Reviewer #1: The authors have addressed all the issues raised. I thank them for their comments and recommend that the manuscript be accepted.

7. PLOS authors have the option to publish the peer review history of their article (what does this mean?). If published, this will include your full peer review and any attached files.

Reviewer #1: **Yes: **Marcela Gomes Reis

---

## [Editor Report · Acceptance letter]

21 Dec 2024

PONE-D-24-43823R3 

PLOS ONE

Dear Dr. Kossmann, 

I'm pleased to inform you that your manuscript has been deemed suitable for publication in PLOS ONE. Congratulations! Your manuscript is now being handed over to our production team.

Kind regards, 

on behalf of

Dr. António Raposo 

Academic Editor

PLOS ONE